# Disentangling Token Dependencies for Efficient Decoding in Diffusion Language Models

## Abstract

Diffusion-based large language models (dLLMs) generate text by gradually filling in masked tokens. However, they're still slow because they usually decode only one or a few tokens per step. Parallel decoding, which unmasks multiple tokens simultaneously, offers a promising way to accelerate generation, but it often degrades output quality when too many tokens are predicted at once. We identify the root cause: unnecessary dependencies between decoded tokens. When multiple tokens are decoded together, the model may incorrectly condition predictions on each other rather than relying solely on the already-generated context. This leads to reduced output quality. To address this, we propose **Disentangled Decoding**, a training–inference framework that suppresses harmful intra-step dependencies in dLLM parallel decoding. *In training*, we introduce dependency-aware self-distillation. The model learns, in a single forward pass, to reproduce what a sequential two-step decoding would produce. This encourages the model to predict multiple tokens based solely on global context rather than jointly decoded tokens. *At inference*, we introduce Slow-Fast Decoding, a dynamic strategy that tailors parallelism to each token's dependency on context. We quantify this dependency using Jensen–Shannon Divergence (JSD). Tokens that are highly dependent on the already-generated context are grouped for faster parallel generation; Other tokens are decoded slowly. Together, these components enable stable, high-quality generation of up to five tokens per step. Across four benchmarks, our method achieves up to $3.3\times$ speedup over vanilla greedy decoding, with minimal loss in generation quality. Please see our project page at https://anonymous.4open.science/r/dsquare-dlm.

## 1 Introduction

Generative models for natural language have become thehe cornerstone of modern artificial intelligence, enabling a vast array of applications. Among these, Masked Diffusion Models (MDMs) (Nie et al., 2025b; Ye et al., 2025) have emerged as a powerful and promising paradigm. By iteratively denoising a sequence from a fully masked state, MDMs offer a highly parallelizable framework for generation. This inherent parallelism presents a significant advantage, holding the potential for substantial improvements in generation speed and efficiency, a critical factor for the deployment of large-scale language models in real-world scenarios.

In practice, however, this potential for speed remains underutilized because most MDMs decode only a few tokens at each step. Typically, the sequence is divided into blocks, and within each block, tokens are revealed incrementally over multiple steps. Confidence-aware parallel decoding (Wu et al., 2025; Yu et al., 2025) accelerates this by unmasking all tokens whose predicted probability exceeds a high threshold. Yet, pushing for greater speed, by lowering the threshold to decode more tokens per step, invariably leads to a sharp drop in generation quality. This sharp speed–quality trade-off suggests that current models are not truly ready for aggressive parallelism.

We identify the root cause: unnecessary dependencies between tokens decoded in the same step. To analyze this, we introduce the perspective of viewing MDM decoding as an *iterative token grouping* process, where the goal at each step is to identify the largest possible group of tokens that can be predicted in parallel without sacrificing coherence. The performance degradation at low confidence thresholds occurs when this grouping is suboptimal, forcing tokens with strong

inter-dependencies to be decoded simultaneously, thereby violating the underlying conditional independence assumption. Our central hypothesis is that standard training objectives cause models to learn *unnecessary dependencies*—spurious or overly rigid correlations between tokens that are not linguistically essential but create artificial computational bottlenecks.

To address this, we propose Disentangled Decoding, a unified framework that tackles the problem at both training and inference time. Our goal is to eliminate harmful intra-step dependencies.

In training, we introduce Dependency-Aware Self-Distillation. Specifically, the model is re-trained to reproduce, in one forward pass, what a careful model would generate over two sequential decoding steps. This forces the model to predict multiple tokens based on global context alone without artificial couplings while preserving linguistically meaningful structure.

At inference, we complement this with a Slow-Fast Decoding strategy that dynamically partitions tokens based on their sensitivity to the already-generated context. We measure this using Jensen–Shannon Divergence (JSD), which quantifies the difference between its predictive distribution with and without access to the preceding block. Tokens with high JSD are strongly shaped by context and can be safely decoded in parallel; those with low JSD are more ambiguous and decoded slower. This way, we only group together tokens that are truly ready for parallel decoding, balancing speed and quality naturally.

Our contributions are threefold and can be summarized as follows:

- We introduce a novel perspective that frames MDM decoding as an iterative token grouping problem, and identify the learning of unnecessary dependencies as the key bottleneck limiting parallel generation performance.

- We propose Dependency-Aware Self-Distillation, a training method that teaches the model to generate high-quality outputs in one pass by mimicking a two-step sequential decoder, reducing reliance on artificial local dependencies.

- We develop a Slow-Fast Decoding, an inference strategy that uses Jensen–Shannon Divergence to group only those tokens that are truly ready for parallel decoding, preserving quality while accelerating generation.

- Through extensive experiments, we demonstrate that our combined approach significantly pushes the speed-performance frontier for MDMs, achieving substantial acceleration factors with minimal to no loss in generation quality, thereby outperforming existing state-of-the-art methods.

## 2 RELATED WORKS

**Discrete Diffusion Language Models.** Discrete diffusion language models (dLM) have recently been a compelling paradigm for non-autoregressive text generation. Unlike previous left-to-right generation in autoregressive models, these dLM models operate by iteratively refining a sequence from a corrupted state, typically one filled with [MASK] tokens. Pioneering works Gong et al. (2025); Nie et al. (2025a) established the scalability of the masked diffusion language models, demonstrating that these models could effectively leverage large-scale data and parameter counts. With the demonstrated scalability, a series of new powerful diffusion language models Nie et al. (2025b); Ye et al. (2025); Song et al. (2025); Khanna et al. (2025); Zhu et al. (2025) have emerged. Most notably, recent open-source large diffusion language models such as LLaDA Nie et al. (2025b) and Dream Ye et al. (2025) have achieved performance that is highly competitive with autoregressive counterparts of comparable model scales, underscoring their viability as a promising architecture for generative language tasks. Our work builds upon these works, addressing the critical challenge of inference latency that currently limits their practical deployment.

**Acceleration of Masked Diffusion Models.** Despite their strong performance, a primary challenge for Masked Diffusion Models is their inference latency, which often trails that of highly optimized autoregressive models. This latency stems from two factors. First, the non-autoregressive nature of the decoding process precludes the use of standard KV-caching mechanisms. Several works have proposed specialized caching variants to reduce redundant computations in this setting Ma et al. (2025); Liu et al. (2025); Wu et al. (2025); Wang et al. (2025). Second, and more central to our work, is the bottleneck within the iterative decoding process itself. Previous approaches Nie et al. (2025b);

Ye et al. (2025) often employ a greedy decoding strategy, decoding only the single most confident token per step, which is computationally inefficient. Confidence-aware parallel decoding Wu et al. (2025) mitigates this by simultaneously unmasking all tokens whose predicted confidence exceeds a high threshold. However, this approach is constrained by a sharp speed-performance trade-off: lowering the threshold to increase parallelism and accelerate inference invariably leads to a significant degradation in generation quality. Our work explores to tackle this challenge to enabling it to confidently generate larger groups of tokens per step by reshaping its learned tokens dependencies through self-distillation and grouping constrain.

## 3 MASKED DIFFUSION MODELS DECODING AS ITERATIVE TOKEN GROUPING

Masked Diffusion Models (MDMs) Nie et al. (2025b); Ye et al. (2025) have recently emerged as a powerful class of generative models for natural language, demonstrating compelling performance on a diverse range of tasks. MDMs operate via a forward noising process that incrementally corrupts an input sequence $\boldsymbol{x}_0$ by replacing its tokens with a special [MASK] token. This process is governed by a predefined noise schedule, and the distribution of a noisy sequence $\boldsymbol{x}_t$ at time $t \in [0, 1]$ conditioned on the original sequence $\boldsymbol{x}_0$ is given by:

$$q(\boldsymbol{x}_t|\boldsymbol{x}_0) = \prod_{i=1}^{n} q(\boldsymbol{x}_t^i|\boldsymbol{x}_0^i) = \prod_{i=1}^{n} \text{Cat}\left(\boldsymbol{x}_t^i; (1-t)\delta_{\boldsymbol{x}_0^i} + t\delta_{[\text{MASK}]}\right). \tag{1}$$

Here, $t$ represents the continuous diffusion time (or noise level), controlling the interpolation between the clean data distribution at $t = 0$ and a fully masked sequence at $t = 1$.

The reverse process, which generates a clean sequence from a fully masked input $\boldsymbol{x}_1$, is learned by a model $p_\theta$. Decoding is typically performed in a semi-autoregressive manner. The sequence is partitioned into $N$ contiguous blocks, $\{B_1, \ldots, B_N\}$. These blocks are generated sequentially. Within each block $B_i$, the masked tokens are denoised over multiple steps. The generation of block $B_i$ is conditioned on the previously generated blocks $\{B_1, \ldots, B_{i-1}\}$ and the still-masked future blocks $\{B_{i+1}, \ldots, B_N\}$:

$$p_\theta(\boldsymbol{x}_{B_i}|\boldsymbol{x}_{B_{<i}}, \boldsymbol{x}_{B_{>i}}^{\text{masked}}) = \prod_{k=1}^{M_i} p_\theta(\boldsymbol{x}_{t_{k-1},B_i}|\boldsymbol{x}_{t_k,B_i}, \boldsymbol{x}_{B_{<i}}, \boldsymbol{x}_{B_{>i}}^{\text{masked}}), \tag{2}$$

where $\boldsymbol{x}_{B_{<i}}$ denotes the set of fully denoised preceding blocks, $\boldsymbol{x}_{B_{>i}}^{\text{masked}}$ denotes the subsequent masked blocks, $1 = t_{M_i} > \cdots > t_1 > t_0 = 0$ is a discrete reverse timestep schedule, and $M_i$ is the number of denoising steps for block $B_i$. A standard greedy approach reveals one token with the highest model confidence at each step, making the number of steps equal to the block length ($M_i = |B_i|$). This sequential intra-block decoding is a significant computational bottleneck.

To mitigate this, confidence-aware parallel decoding strategies Wu et al. (2025); Yu et al. (2025) have been proposed. At each step, all masked tokens with a predicted probability exceeding a certain threshold $\tau$ are decoded simultaneously. If no token's confidence surpasses $\tau$, only the single most confident token is decoded. As theoretically justified Wu et al. (2025), for a high threshold $\tau = 1 - \epsilon$, the predictions for selected tokens are approximately conditionally independent. This allows for parallel decoding that closely approximates the greedy sequential process, achieving significant speedups (e.g., $3\times$) with negligible performance degradation for high $\tau$ (e.g., $\tau = 0.9$).

We argue that this confidence-aware decoding implicitly performs a dynamic token grouping. The key to accelerating MDM decoding lies in minimizing the number of sequential steps, $M_i$, for each block. This is equivalent to finding an optimal partition of the tokens within a block. Let the set of token indices in block $B_i$ be $\mathcal{I}_i$. The decoding process partitions $\mathcal{I}_i$ into an ordered sequence of disjoint groups $\mathcal{P}_i = (G_1, G_2, \ldots, G_{M_i})$, where $\mathcal{I}_i = \bigcup_{k=1}^{M_i} G_k$. The generation of the block can then be expressed as:

$$p_\theta(\boldsymbol{x}_{B_i}|\text{context}) = \prod_{k=1}^{M_i} p_\theta(\boldsymbol{x}_{G_k}|\boldsymbol{x}_{G_{<k}}, \text{context}), \tag{3}$$

where $\boldsymbol{x}_{G_k}$ are the tokens corresponding to indices in group $G_k$, and $\boldsymbol{x}_{G_{<k}}$ are all previously decoded tokens in the block. The parallel decoding strategy makes a crucial conditional independence

assumption within each group:

$$p_\theta(\boldsymbol{x}_{G_k}|\boldsymbol{x}_{G_{<k}}, \text{context}) \approx \prod_{j \in G_k} p_\theta(x_j|\boldsymbol{x}_{G_{<k}}, \text{context}). \tag{4}$$

The number of sequential steps is thus $M_i = |\mathcal{P}_i|$, the number of groups in the partition.

However, a fundamental tension exists. Lowering the confidence threshold $\tau$ reduces $M_i$ by creating larger, more inclusive groups, but it often leads to a sharp decline in generation quality. This performance drop occurs because a lower threshold is more likely to group tokens with strong inter-dependencies into the same step $G_k$. This violates the independence assumption in Eq. 4, causing the model to generate inconsistent or incoherent text.

## 4 METHODOLOGY

We hypothesize that this trade-off is not inherent but is exacerbated by unnecessary dependencies learned by current MDMs. To address this, we propose complementary solutions at both training and inference time.

**Training.** First, in Sec. 4.1, we introduce a self-distillation method designed to regularize the model, removing superfluous dependencies while preserving essential linguistic structures. This enables more aggressive parallel decoding under lower confidence thresholds without sacrificing performance.

**Inference.** Second, we propose Slow-Fast Decoding in Sec 4.2, an inference strategy that dynamically groups tokens based on their sensitivity to already-generated context. As formalized in Eq. 3, we use Jensen–Shannon Divergence (JSD) to measure the dependency between a token's predictive distribution with and without access to the preceding block. Tokens with high JSD are context-stable and decoded in parallel ("fast"); those with low JSD are ambiguous and decoded sequentially ("slow"). This adaptive grouping ensures only compatible tokens are processed together, preserving generation quality while enabling acceleration.

### 4.1 DEPENDENCY-AWARE SELF-DISTILLATION

A primary obstacle to aggressive parallel decoding in MDMs is not linguistic dependency itself, but the model's tendency to learn spurious or overly rigid correlations that create artificial computational bottlenecks. For instance, consider completing the phrase: "The report detailed the company's ______ growth and ______ expansion." Plausible completions could be ("financial", "global"), ("rapid", "market"), or ("steady", "international"). While the words in each pair are semantically related, they are not strictly dependent; the surrounding context strongly supports both tokens independently. However, a standard MDM might learn an overly sensitive conditional model where predicting "global" is difficult until "financial" is revealed. This forces a sequential decoding step that is not linguistically essential—an artifact of an *unnecessary dependency*.

Our goal is to regularize the model to disentangle these unnecessary correlations, encouraging it to rely more on the global context rather than spurious local cues from other masked tokens. This can be formalized by contrasting the probabilistic assumptions of sequential and parallel decoding. For a group of tokens $G$ to be decoded, a cautious teacher model $\theta$ adheres to the chain rule, representing a dependent, sequential generation process:

$$p_\theta(\boldsymbol{x}_G|\text{context}) = \prod_{j=1}^{|G|} p_\theta(x_{g_j}|\boldsymbol{x}_{\{g_1,\dots,g_{j-1}\}}, \text{context}). \tag{5}$$

Conversely, an ideal parallel student model $\theta^+$ would rely on a factorized distribution, assuming conditional independence given the context:

$$p_{\theta^+}(\boldsymbol{x}_G|\text{context}) = \prod_{j=1}^{|G|} p_{\theta^+}(x_{g_j}|\text{context}). \tag{6}$$

Our objective is to make the student's parallel model (Eq. 6) a high-fidelity approximation of the teacher's more robust, sequential generation (Eq. 5), specifically for token groups where the independence assumption is linguistically plausible.

To achieve this, we introduce *dependency-aware self-distillation*. The process requires training data that faithfully mirrors the semi-autoregressive inference setting. For a given sequence $\boldsymbol{x}_0$, we create an input $\boldsymbol{x}_t$ by randomly selecting a block $B_i$, leaving preceding blocks $B_{<i}$ clean, masking subsequent blocks $B_{>i}$, and applying noise at a random level $t \in [0, 1]$ to the active block $B_i$.

The distillation process trains a student model $\theta^+$ using a frozen, identical teacher model $\theta$. Given an input $\boldsymbol{x}_t$ with masked indices $\mathcal{M}_t$, we derive a sophisticated target distribution from the teacher in a two-step process.

**Teacher's Two-Step Target Generation.** First, the teacher performs an initial pass to compute logits $\boldsymbol{z}^{(1)} = f_\theta(\boldsymbol{x}_t)$ and identifies a set of "independently plausible" tokens $\mathcal{K} = \{k \in \mathcal{M}_t \mid \max_v \sigma(\boldsymbol{z}^{(1)})_k^v > \tau_{\text{tr}}\}$, where $\sigma$ is the softmax function. These tokens are decoded to form a more clean sequence $\boldsymbol{x}_s$. Second, the teacher performs a refined pass $f_\theta(\boldsymbol{x}_s)$ to obtain updated logits $\boldsymbol{z}^{(2)}$ for the remaining, more ambiguous and dependent tokens. The final target logits $\hat{\boldsymbol{z}}$ are a composite, using the original predictions for the confident set and the refined predictions for the rest:

$$\hat{\boldsymbol{z}}_k = \begin{cases} \boldsymbol{z}_k^{(1)} & \text{if } k \in \mathcal{K} \\ \boldsymbol{z}_k^{(2)} & \text{if } k \in \mathcal{M}_t \setminus \mathcal{K} \end{cases} \quad \forall k \in \mathcal{M}_t. \tag{7}$$

This target encapsulates the teacher's belief after a careful, sequential reasoning step.

**Student Training and Objective.** The student model performs only a single forward pass on the initial input $\boldsymbol{x}_t$ to produce its logits $\boldsymbol{z}^+ = f_{\theta^+}(\boldsymbol{x}_t)$. We align the student with the teacher's composite target by minimizing the KL divergence between their output distributions over all initially masked tokens. The loss is weighted by the inverse of the sequence-level noise ratio $\hat{t}$ (the total fraction of masked tokens in $\boldsymbol{x}_t$):

$$\mathcal{L}_{\text{distill}} = \frac{1}{\hat{t}} \mathbb{E}_{\boldsymbol{x}_t \sim q(\boldsymbol{x}_t|\boldsymbol{x}_0)} \left[ \sum_{k \in \mathcal{M}_t} \text{KL}\left( \sigma(\hat{\boldsymbol{z}}_k) \,\big\|\, \sigma(\boldsymbol{z}_k^+) \right) \right]. \tag{8}$$

By minimizing this objective, the student learns to directly produce the teacher's refined output in one step. It is explicitly trained to co-predict the tokens in $\mathcal{K}$ in parallel, effectively pruning the unnecessary dependencies that would have otherwise forced a sequential generation, while preserving the necessary conditional reasoning for more complex tokens.

---

**Algorithm 1:** Dependency Aware Self Distillation

**Input:** Frozen teacher $\theta$, student $\theta^+$, sequence $\boldsymbol{x}_0$, confidence threshold $\tau_{\text{tr}}$
**Output:** Updated student parameters $\theta^+$
**for** *each training iteration* **do**

    Sample an active block index $i$ and a noise level $t \sim \mathcal{U}(0, 1)$;
    Construct $\boldsymbol{x}_t$ by keeping $B_{<i}$ clean, masking $B_{>i}$, and applying noise of level $t$ to $B_i$;
    Let $\mathcal{M}_t$ be the set of masked indices and set $\hat{t} \leftarrow |\mathcal{M}_t|/|\boldsymbol{x}_0|$;
    Decode tokens set $\mathcal{K} \leftarrow \left\{ k \in \mathcal{M}_t \mid \max_v \pi_k^{(1)v} > \tau_{\text{tr}} \right\}$, where $\pi^{(1)} \leftarrow \sigma(\boldsymbol{z}^{(1)})$;
    Form $\boldsymbol{x}_s$ by decoding tokens at $\mathcal{K}$ with $\arg\max_v \pi_k^{(1)v}$;
    Sample one more step with the teacher model $\boldsymbol{z}^{(2)} \leftarrow f_\theta(\boldsymbol{x}_s)$;
    Composite teacher target **for** $k \in \mathcal{M}_t$ **do**
        $\hat{\boldsymbol{z}}_k \leftarrow \begin{cases} \boldsymbol{z}_k^{(1)}, & k \in \mathcal{K} \\ \boldsymbol{z}_k^{(2)}, & k \in \mathcal{M}_t \setminus \mathcal{K} \end{cases}$;
    Forward student model with the input of $\boldsymbol{x}_t$, $\boldsymbol{z}^+ \leftarrow f_{\theta^+}(\boldsymbol{x}_t)$;
    Employ the KLD loss on all masked tokens $\mathcal{L}_{\text{distill}} \leftarrow \frac{1}{\hat{t}} \sum_{k \in \mathcal{M}_t} \text{KL}\left( \sigma(\hat{\boldsymbol{z}}_k) \,\big\|\, \sigma(\boldsymbol{z}_k^+) \right)$;
    Update $\theta^+$ by one gradient step to minimize $\mathcal{L}_{\text{distill}}$;

---

## 4.2 SLOW-FAST DECODING BASED ON JSD

While our dependency-aware self-distillation method (Sec. 4.1) effectively prunes unnecessary dependencies, the confidence score remains an imperfect proxy for the true conditional independence

required for parallel decoding. Especially at lower confidence thresholds, tokens with strong, yet-unresolved dependencies can be erroneously grouped together, leading to a degradation in generation quality. To mitigate this risk, we introduce a complementary mechanism: a JSD-based constraint that provides a more direct measure of contextual dependency to guide the token grouping process.

Our key insight is that within any given block $B_i$, the uncertainty of a masked token is influenced by two primary sources: the already-generated context from previous blocks ($\boldsymbol{x}_{B_{<i}}$), and the yet-to-be-generated context from other masked tokens within the same block. Tokens whose resolution is highly dependent on the previous blocks are critical "linchpin" tokens; their incorrect generation can derail the entire sequence. Therefore, we can quantify context dependency of tokens using the Jensen-Shannon Divergence (JSD), which measures the difference between a token's predictive distribution with and without access to the denoised previous block.

Formally, for each masked token $j$ in the active block $B_i$, we compute its token-wise JSD as:

$$\mathcal{J}_j = \text{JSD}\left(p_\theta(\cdot|\boldsymbol{x}_{B_{<i}}, \boldsymbol{x}_{B_i \setminus \{j\}}^{\text{masked}}, \dots) \,\middle\|\, p_\theta(\cdot|\boldsymbol{x}_{B_{<i}}^{\text{masked}}, \boldsymbol{x}_{B_i \setminus \{j\}}^{\text{masked}}, \dots)\right), \quad (9)$$

where $p_\theta(\cdot|\text{context})$ is the model's predicted probability distribution for token $j$. A high $\mathcal{J}_j$ indicates that the model's prediction for token $j$ changes significantly once the prior context $\boldsymbol{x}_{B_{<i}}$ is revealed, marking it as highly dependent on that context. Conversely, a low $\mathcal{J}_j$ suggests the token is relatively stable and primarily constrained by the global structure of the sentence rather than the specific preceding words.

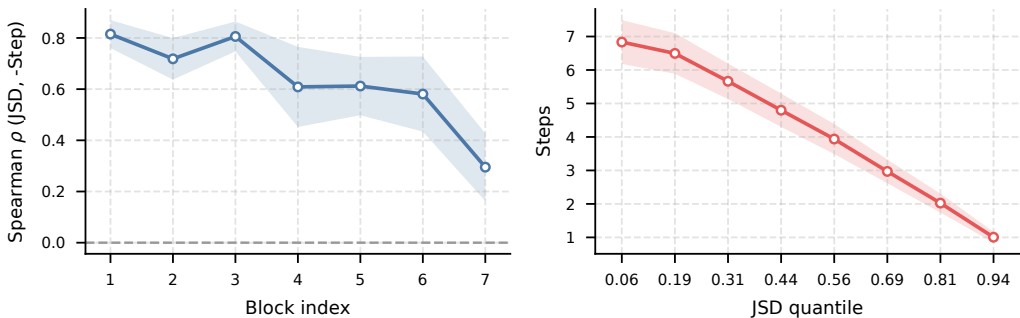

Figure 1: Blockwise relationship between JSD and decoding order. We aggregate tokens from blocks 1–7 (block 0 excluded) across all samples. Steps denotes the decoding iteration for a specific token in one block. (a) Mean Spearman $\rho(JSD, -steps)$ per block with $95\%$ confidence interval (shaded). Positive $\rho$ indicates that higher-JSD tokens tend to be decoded earlier within the block. (b) Mean Steps versus JSD quantile (0–1; rank-based bins), with $95\%$ confidence interval. The decreasing curve shows that tokens with larger JSD are decoded earlier relative to peer tokens in the same block.

As empirical statistics shown in Fig. 2, we analyze the block-wise relationship between JSD and token decoding order. These results motivate the design of our decoding strategy by revealing that tokens with higher JSD tend to be decoded earlier within a block.

Building on this observation, we leverage the JSD metric to implement a dynamic, hybrid decoding strategy. Instead of using a single, low confidence threshold $\tau_{\text{low}}$, we partition the masked tokens in block $B_i$ into two sets based on their JSD scores. A fixed, absolute JSD threshold would be brittle and context-agnostic. Therefore, we propose a more robust, adaptive threshold based on the distribution of JSD scores within the block itself. Specifically, we define a "slow set" $\mathcal{S}_{\text{slow}}$ and a "fast set" $\mathcal{S}_{\text{fast}}$:

$$\mathcal{S}_{\text{slow}} = \{j \in B_i \mid \mathcal{J}_j > \text{mean}(\{\mathcal{J}_k\}_{k \in B_i})\}, \quad \mathcal{S}_{\text{fast}} = B_i \setminus \mathcal{S}_{\text{slow}}. \quad (10)$$

Tokens in the fast set (low JSD) are decoded using an aggressive low confidence threshold $\tau_{\text{low}}$, permitting high parallelism. Tokens in the slow set (high JSD), being more critical and context-dependent, are decoded using a conservative high threshold $\tau_{\text{high}}$ until all have been revealed. This hybrid approach allows for rapid decoding of stable tokens while ensuring careful, sequential treatment of pivotal ones.

The effectiveness of this JSD-based partitioning is not merely empirical; it is grounded in the goal of minimizing the error introduced by the parallel decoding assumption. We formalize this in the *Appendix* Sec. A.2.

## 5 EXPERIMENTS

### 5.1 EXPERIMENTS SETUP

Our experiments are conducted using the representative masked diffusion language model LLaDA-8B-Instruct Nie et al. (2025b). For dependency-aware self distillation, the training data is generated with LLaDA-8B-Instruct model on the GSM8K Cobbe et al. (2021) training split with a sequence length of $1,024$ and block length of $128$, resulting in a total of 7.3K paired training samples. All sequences are pre-filtered and truncated to the maximum length of $1,024$ to ensure consistency across samples. o minimize distributional shift during fine-tuning, we employ Low-Rank Adaptation (LoRA) with rank 32, scaling factor 32, and a dropout rate of $0.1$. The confidence threshold for selecting independent tokens in the teacher's first decoding step is set to $\tau_{\mathrm{tr}} = 0.98$. Both training and inference are performed with a fixed block size of 32 tokens to keep consistency..

For the JSD-based constraint applied during inference, we fix the confidence threshold for the "slow" set at $\tau_{\mathrm{high}} = 0.9$, while the threshold for the "fast" set, denoted $\tau_{\mathrm{low}}$, is adjustable depending on the desired decoding speed. During full-sequence generation, decoding starts with $\tau_{\mathrm{low}}$ applied to all tokens. The JSD-based partitioning is activated beginning from the second block, as it requires computing the JSD between the preceding block $B_{i-1}$ and the current block $B_i$. Tokens in $B_i$ are dynamically assigned to either the "fast" or "slow" set based on their JSD scores. After more than $60\%$ of the "fast" tokens in $B_i$ have been decoded, the remaining tokens, including those in the "slow" set, are also decoded using the lower threshold $\tau_{\mathrm{low}}$ to continue parallel generation efficiently.

### 5.2 MAIN RESULTS AND ANALYSIS

**Evaluation Benchmarks.** Following common evaluation protocols, we evaluate our method on four representative benchmarks spanning mathematical reasoning and code generation: GSM8K Cobbe et al. (2021), HumanEval Chen et al. (2021), MATH Lewkowycz et al. (2022), and MBPP Austin et al. (2021). These benchmarks are widely adopted to assess both the reasoning capability and generation accuracy of large language models.

| Benchmark | Baseline Greedy | Fast-dLLM | | | Self-Distillation | | |
|---|---|---|---|---|---|---|---|
| | | $\tau = 0.9$ | $\tau = 0.8$ | $\tau = 0.7$ | $\tau = 0.9$ | $\tau = 0.8$ | $\tau = 0.7$ |
| GSM8K (5-shot) | 79.3 | 78.8 | 77.7 | 76.2 | 78.9 | 79.2 | 78.6 |
| | 5.2 | 12.7 ( 2.5× ) | 16.2 ( 3.1× ) | 20.3 ( 3.9× ) | 11.01 ( 2.1× ) | 14.1 ( 2.7× ) | 17.6 ( 3.4× ) |
| MATH (4-shot) | 33.5 | 33.6 | 33.1 | 31.8 | 33.5 | 32.7 | 31.9 |
| | 7.0 | 9.1( 1.3× ) | 9.8( 1.4× ) | 11.90 ( 1.7× ) | 12.7 ( 2.7× ) | 15.9 ( 3.2× ) | 19.4 ( 3.8× ) |
| HumanEval (0-shot) | 41.5 | 42.7 | 38.4 | 34.1 | 39.6 | 37.8 | 32.9 |
| | 16.3 | 54.3 ( 3.3× ) | 67.3 ( 4.1× ) | 81.12 ( 5.0× ) | 48.7 ( 3.0× ) | 60.1 ( 3.7× ) | 73.2 ( 4.5× ) |
| MBPP (3-shot) | 29.4 | 29.6 | 29.2 | 26.4 | 29.2 | 29.0 | 26.8 |
| | 3.3 | 13.6 ( 4.1× ) | 16.7 ( 5.1× ) | 20.2 ( 6.1× ) | 12.2 ( 3.7× ) | 14.8 ( 4.5× ) | 17.7 ( 5.4× ) |

Table 1: Benchmark results on LLaDA-8B-Instruct with self-distillation only. To compare with Fast-dLLM Wu et al. (2025), we gradually lower confidence threshold in decoding from 0.9 to 0.7.

| Benchmark | Baseline Greedy | Fast-dLLM | | | Self-Distillation with JSD constrain | | |
|---|---|---|---|---|---|---|---|
| | | $\tau = 0.9$ | $\tau = 0.8$ | $\tau = 0.7$ | $\tau = 0.9$ | $\tau = 0.8$ | $\tau = 0.7$ |
| GSM8K (5-shot) | 79.3 | 78.8 | 77.7 | 76.2 | 78.9 | 78.8 | 79.2 |
| | 5.2 | 12.7 ( 2.5× ) | 16.2 ( 3.1× ) | 20.3 ( 3.9× ) | 11.01 ( 2.1× ) | 13.8 ( 2.7× ) | 17.1 ( 3.3× ) |
| HumanEval (0-shot) | 41.5 | 42.7 | 38.4 | 34.1 | 39.6 | 37.2 | 33.0 |
| | 16.3 | 54.3 ( 3.3× ) | 67.3 ( 4.1× ) | 81.12 ( 5.0× ) | 48.7 ( 3.0× ) | 59.6 ( 3.7× ) | 72.1 ( 4.4× ) |
| MBPP (3-shot) | 29.4 | 29.6 | 29.2 | 26.4 | 29.2 | 29.2 | 26.9 |
| | 3.3 | 13.6 ( 4.1× ) | 16.7 ( 5.1× ) | 20.2 ( 6.1× ) | 12.2 ( 3.7× ) | 14.7 ( 4.5× ) | 17.3 ( 5.2× ) |

Table 2: Benchmark results on LLaDA-8B-Instruct with self-distillation and JSD-based constrain.

**Comparison with Baselines.** We compare our results against two baselines: greedy decoding from LLaDA Nie et al. (2025b) and parallel decoding with a fixed threshold $\tau = 0.9$ from Fast-dLLM Wu et al. (2025). All evaluations and speed measurements are conducted on an NVIDIA A4500 GPU.

We first present results using only our proposed self-distillation method in Table 1. The results demonstrate that our method consistently achieves higher accuracy on GSM8K compared to Fast-dLLM under equivalent confidence thresholds. Although performance on HumanEval and MBPP shows a slight decline, this is primarily due to the self-distillation being conducted exclusively on a mathematical reasoning dataset. To mitigate this limitation, we extend our distillation training to other domains, and those results are reported in the *Appendix*.

Overall experimental results on GSM8K, HumanEval, and MBPP are summarized in Table 2. Our model, enhanced with self-distillation and JSD-based decoding constraint, achieves an accuracy of 79.2 on GSM8K with a $3.3\times$ speedup relative to greedy decoding. Under the same confidence threshold, our method consistently outperforms Fast-dLLM Wu et al. (2025) on GSM8K. Moreover, on both MBPP and HumanEval benchmarks, our approach yields consistent accuracy improvements, particularly in decoding with low threshold. When compared to Table 1, the integration of the JSD constraint introduces only negligible computational overhead. Specifically, at $\tau = 0.7$ on the GSM8K benchmark, the decoding speed decreases only slightly from 17.6 to 17.1 tokens per second.

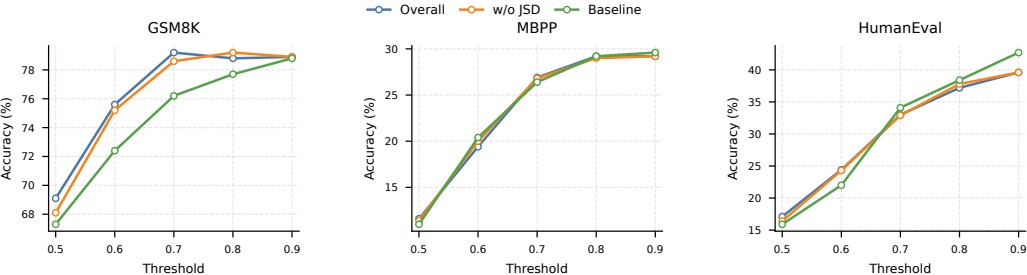

Figure 2: Accuracy trends on GSM8K, MBPP, and HumanEval as the confidence threshold is lowered from 0.9 to 0.5.

## 6 LIMITATION

While our proposed Disentangled Decoding framework significantly improves the speed-quality trade-off in Masked Diffusion Models, it is not without limitations. First, our self-distillation approach relies on synthetic training data generated by teacher model, which may introduce domain bias when generalizing to out-of-distribution tasks, such as code generation or open-domain dialogue. Although our method demonstrates strong performance on mathematical reasoning benchmarks, its transferability to broader domains may require additional task-specific distillation data. Second, the computation of Jensen–Shannon Divergence (JSD) during inference introduces modest overhead, especially in early decoding stages where accurate context modeling is most critical. While this overhead is minimal relative to the performance gains, it may still pose a bottleneck in extremely latency-sensitive deployment scenarios. Finally, our current design assumes a fixed block structure and uniform token partitioning, which may not optimally align with the dynamic nature of linguistic dependencies. Future work could explore adaptive block scheduling or hierarchical grouping mechanisms to further enhance decoding flexibility.

## 7 CONCLUSION

This work presents *Disentangled Decoding*, a unified framework for improving the efficiency and robustness of Masked Diffusion Models (MDMs) through targeted mitigation of unnecessary token dependencies. By viewing parallel decoding as an iterative token grouping problem, we identify over-learned intra-step dependencies as a key barrier to speed-quality trade-offs. Our proposed Dependency-Aware Self-Distillation enables the model to internalize cleaner, context-based predictions during training, while the JSD-based grouping constraint adaptively regulates token selection at inference time. Extensive evaluations across mathematical reasoning and code generation tasks demonstrate that our approach significantly enhances decoding speed—achieving up to $3.3\times$ acceleration without compromising generation quality. These results establish a promising direction for making MDMs truly scalable in real-world applications requiring fast, high-quality language generation.

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

# A APPENDIX

## A.1 THE USE OF LARGE LANGUAGE MODELS (LLMS)

The research ideas and experimental design of this paper were conceived entirely by the authors without the use of LLMs. During manuscript preparation, we used GPT to assist with grammar checking and refinement of language for clarity and readability.

## A.2 THEORETICAL JUSTIFICATION FOR THE JSD-BASED CONSTRAINT

**Theorem 1.** *Let the decoding process for a block $B_i$ be a partition into an ordered sequence of groups $\mathcal{P}_i = (G_1, \ldots, G_M)$. The error incurred at step $k$ due to the parallel decoding assumption is given by the KL divergence between the true sequential joint and the factorized approximation:*

$$E_{G_k} = \mathrm{KL}\left(p_\theta(\boldsymbol{x}_{G_k}|\mathcal{C}_k) \;\middle\|\; \prod_{j \in G_k} p_\theta(x_j|\mathcal{C}_k)\right), \tag{11}$$

*where $\mathcal{C}_k = (\boldsymbol{x}_{B_{<i}}, \boldsymbol{x}_{G_{<k}}, \ldots)$ is the full context available before decoding group $G_k$. The total error for the block is $E_{total} = \sum_{k=1}^{M} E_{G_k}$.*

*The JSD score for a token $j \in B_i$, defined as $\mathcal{J}_j = \mathrm{JSD}\left(p_\theta(\cdot|\mathcal{C}_1) \,\|\, p_\theta(\cdot|\mathcal{C}_0)\right)$, where $\mathcal{C}_1$ is the context with the true past block $\boldsymbol{x}_{B_{<i}}$ and $\mathcal{C}_0$ is the context with it masked, quantifies the token's sensitivity to past-block context. A decoding strategy that applies a more conservative grouping (i.e., smaller group sizes) to tokens with higher $\mathcal{J}_j$ scores serves as a principled approach to minimizing the total expected generation error $E_{total}$.*

*Proof.* The proof proceeds in three parts. First, we decompose the group error term $E_{G_k}$ to reveal its dependence on intra-group conditional information. Second, we relate the JSD metric to information-theoretic quantities that measure contextual sensitivity. Finally, we argue that high contextual sensitivity, as measured by JSD, implies a higher expected contribution to the error term, justifying the proposed constrained grouping strategy.

**1. Decomposing the Parallelization Error.** The error term $E_{G_k}$ quantifies the discrepancy introduced by ignoring the dependencies among tokens within the group $G_k$. Using the chain rule for probability on the true joint, $p_\theta(\boldsymbol{x}_{G_k}|\mathcal{C}_k) = \prod_{j \in G_k} p_\theta(x_j|\boldsymbol{x}_{G_k, <j}, \mathcal{C}_k)$, where $<j$ denotes an arbitrary but fixed ordering within the group. The KL divergence can be expanded as follows:

$$E_{G_k} = \mathbb{E}_{\boldsymbol{x}_{G_k} \sim p_\theta(\cdot|\mathcal{C}_k)}\left[\log p_\theta(\boldsymbol{x}_{G_k}|\mathcal{C}_k) - \log \prod_{j \in G_k} p_\theta(x_j|\mathcal{C}_k)\right] \tag{12}$$

$$= \mathbb{E}_{\boldsymbol{x}_{G_k} \sim p_\theta(\cdot|\mathcal{C}_k)}\left[\sum_{j \in G_k} \log p_\theta(x_j|\boldsymbol{x}_{G_k, <j}, \mathcal{C}_k) - \sum_{j \in G_k} \log p_\theta(x_j|\mathcal{C}_k)\right] \tag{13}$$

$$= \sum_{j \in G_k} \mathbb{E}_{\boldsymbol{x}_{G_k, \leq j} \sim p_\theta(\cdot|\mathcal{C}_k)}\left[\log \frac{p_\theta(x_j|\boldsymbol{x}_{G_k, <j}, \mathcal{C}_k)}{p_\theta(x_j|\mathcal{C}_k)}\right] \tag{14}$$

$$= \sum_{j \in G_k} \mathbb{E}_{\boldsymbol{x}_{G_k, <j} \sim p_\theta(\cdot|\mathcal{C}_k)}\left[\mathrm{KL}\left(p_\theta(\cdot|\boldsymbol{x}_{G_k, <j}, \mathcal{C}_k) \,\|\, p_\theta(\cdot|\mathcal{C}_k)\right)\right]. \tag{15}$$

This decomposition shows that the total error for a group is the sum of expected KL divergences. Each term represents the information gained about a token $x_j$ from knowing the other tokens decoded just before it within the same parallel step. The parallel decoding error is large if tokens within a group strongly inform one another.

**2. The JSD as a Measure of Contextual Sensitivity.** The Jensen-Shannon Divergence between the predictive distributions for token $j$ under context $\mathcal{C}_1$ (past revealed) and $\mathcal{C}_0$ (past masked) is defined as:

$$\mathcal{J}_j = \frac{1}{2}\mathrm{KL}(p_1\|p_M) + \frac{1}{2}\mathrm{KL}(p_0\|p_M), \tag{16}$$

where $p_1 = p_\theta(\cdot|\mathcal{C}_1)$, $p_0 = p_\theta(\cdot|\mathcal{C}_0)$, and $p_M = \frac{1}{2}(p_1 + p_0)$ is the mixture distribution. The JSD is the mutual information between the random variable for token identity $X_j$ and a binary random variable $C$ representing the context choice ($C = 0$ for $\mathcal{C}_0$, $C = 1$ for $\mathcal{C}_1$). A high $\mathcal{J}_j$ signifies that revealing the past context provides substantial information about the identity of token $j$, implying that the token's predictive distribution is highly sensitive to its surrounding context. Such tokens are often linguistically pivotal, resolving significant ambiguity in the sequence.

**3. Linking Contextual Sensitivity to Parallelization Error.** The core of our argument rests on the well-founded linguistic assumption that a token's sensitivity to its context is a general property. A token whose identity is highly uncertain without the preceding block's context (high $\mathcal{J}_j$) is also likely to be one whose identity is highly uncertain without the context provided by its peer tokens within a decoding group. This is because both contexts serve to resolve ambiguity.

Let us consider a token $j$ with a high JSD score, $\mathcal{J}_j$. This indicates that its predictive distribution $p_\theta(x_j|\cdot)$ is highly variable with respect to changes in the conditioning context. When such a token is placed in a large parallel group $G_k$, it is plausible that the information provided by its peer tokens $\boldsymbol{x}_{G_k, <j}$ would also cause a significant shift in its distribution. This leads to a large value for the corresponding KL term in the error decomposition (Eq. 15).

$$\mathbb{E}_{\boldsymbol{x}_{G_k,<j}} \left[ \mathrm{KL}\left(p_\theta(\cdot \mid \boldsymbol{x}_{G_k,<j}, \mathcal{C}_k) \;\|\; p_\theta(\cdot \mid \mathcal{C}_k)\right) \right] \text{ is expected to be large if } \mathcal{J}_j \text{ is large.} \quad (17)$$

Consequently, including tokens with high JSD scores in large parallel groups is likely to contribute disproportionately to the total generation error $E_{\text{total}}$.

Our proposed strategy directly mitigates this risk. By partitioning tokens into a "slow set" $\mathcal{S}_{\text{slow}}$ (high JSD) and a "fast set" $\mathcal{S}_{\text{fast}}$ (low JSD), we isolate the high-risk tokens. Applying a conservative decoding strategy (e.g., high confidence threshold $\tau_{\text{high}}$, leading to small or singleton groups) to $\mathcal{S}_{\text{slow}}$ ensures that these sensitive tokens are decoded with more complete context, thereby minimizing their contribution to the parallelization error. Conversely, for tokens in $\mathcal{S}_{\text{fast}}$, their low JSD suggests robustness to contextual variations, making the factorized approximation in Eq. 6 more accurate and justifying an aggressive parallelization strategy. This hybrid approach thus provides a principled method for managing the speed-quality trade-off by allocating computational caution where it is most needed, thereby minimizing the total expected error. $\qquad\square$

