# OpenReview forum: "Disentangling Token Dependencies for Efficient Decoding in Diffusion Language Models"
_ICLR.cc/2026/Conference — ICLR 2026 Conference Withdrawn Submission_

### Official Review · Reviewer_sRpe · 2025-10-30

**Soundness:** 2
**Presentation:** 3
**Contribution:** 2
**Rating:** 2
**Confidence:** 3

**Summary:**

The paper proposes a framework to speed up diffusion-based language models. It introduces a self distillation mechanism, where the student learns to match a teacher’s two-step decoding in one pass. It also proposes an adaptive decoding scheme, where the tokens are decoded based on their context dependence. The empirical evaluation shows somewhat promising results.

**Strengths:**

Timely and relevant topic: The paper addresses decoding inefficiency in diffusion-based language models, a problem that is becoming increasingly important as these models gain popularity.

Clear and well-structured writing: The exposition is polished, with a logical flow and well-presented figures and tables that make the method easy to follow.

Some empirical gains: The method demonstrates measurable speedups (up to ~3x over greedy decoding) with limited quality degradation, showing partial practical promise.

**Weaknesses:**

Theoretical motivation. The work claims that standard training objectives cause models to learn unnecessary dependencies. I am not fully convinced by this. Why is this so? Under correct maximum-likelihood training (and a powerful enough model), the model should recover the true conditional dependencies in the data. If decoding inefficiency arises, it could reflect properties of the data, and not “spurious” dependencies introduced by training. Am I missing something?

Weak empirical gains? The reported “3.3x speedup” is measured against a one-token-at-a-time greedy diffusion baseline, not against the stronger Fast-dLLM decoder, which already achieves similar acceleration to the approach proposed in the paper. The actual incremental improvement over Fast-dLLM appears to be quite small, and fast dLLM outperforms the method in several tasks. Related, the claim “significantly improves the speed-quality trade-off in Masked Diffusion Models” appears to be an overstatement? Am I misreading results?

Baselines (related to the point above). The greedy baseline decodes one token at a time, while Fast-dLLM combines both parallel decoding and KV caching. Since the proposed method only addresses token dependencies (not caching), it would be more informative to compare directly against Fast-dLLM using parallel decoding without KV caching, to isolate the impact of the proposed dependency disentanglement. Without this, it is difficult to attribute the reported speedups to the new method rather than to general parallel decoding effects.

Clarity of sampling algorithm. Table 1 reports results “with self-distillation only”. It is unclear to me whether confidence-based sampling is still applied during decoding. Since Fast-dLLM relies on confidence thresholds for parallel decoding, the lack of clarity makes it difficult to disentangle the contributions of the proposed self-distillation from those of standard confidence-based sampling.

Impact of the training objective. I have some questions about the two-stage training objective. When the teacher’s confidence threshold is high, the two-step distillation loss likely reduces to standard training (since the very confident tokens are sampled pretty much deterministically), providing little benefit. Conversely, a low threshold risks mode dropping by forcing the student toward overconfident modes of the teacher’s distribution.

**Questions:**

The tables do not explain the metrics reported. I understand the values in black are accuracy, and the blue numbers speedups (tokens per second)? A brief comment could be included in the captions.

Overall, I see the work combines several elements: self-distillation, JSD-based dependency scoring, and confidence-based decoding. I think more ablations targeting the individual effects of each of these components would be useful. It remains unclear which component contributes most to the reported improvements, or whether the gains could be achieved through simpler design choices.

---

### Official Review · Reviewer_bzTL · 2025-11-01

**Soundness:** 3
**Presentation:** 3
**Contribution:** 2
**Rating:** 4
**Confidence:** 4

**Summary:**

The paper presents an interesting approach to improving the decoding process in Masked Diffusion Models by addressing the root cause of performance degradation: unnecessary dependencies between decoded tokens. The method is well-motivated and introduces a dual-layered solution: self-distillation during training and dynamic decoding strategies at inference, which balances the trade-off between speed and quality. Experimental results show notable improvements in generation speed while maintaining quality. However, there are a few concerns regarding the potential limitations of domain generalizability and the computational overhead introduced by the JSD-based strategy. Overall, the proposed framework is a significant step forward in improving the practical usability of MDMs.

**Strengths:**

1. The self-distillation method proposed effectively reduces artificial token dependencies, making the model more robust during parallel decoding.
2. Disentangled Decoding, offers a fresh perspective on how to structure token generation in MDMs for enhanced performance.
3. The use of JSD to guide token grouping based on dependency strength is a novel and technically sound approach that ensures only context-stable tokens are decoded quickly.

**Weaknesses:**

1. The authors claim their method overcomes the speed-performance trade-off of Confidence-aware parallel decoding in Fast-dLLM by reshaping token dependencies through self-distillation and grouping constraints. However, the experimental results do not show a clear improvement compared to Fast-dLLM.
2. While the paper mentions extending training to other domains, the current focus on mathematical and code tasks does not fully demonstrate the method’s versatility. More validation in areas such as language comprehension and logical reasoning would be beneficial.
3.  I'd like to suggest adopting an adaptive block structure during decoding instead of a fixed partitioning. This would better handle linguistic dependencies across different text types. At a minimum, the scalability of the method should be validated by testing it with different block sizes.
4. The experiments used only LLaDA-8B-Instruct. Testing the method with different models would help assess its generalizability and robustness.

**Questions:**

see weaknesses.

---

### Official Review · Reviewer_95fm · 2025-11-02

**Soundness:** 3
**Presentation:** 2
**Contribution:** 1
**Rating:** 2
**Confidence:** 3

**Summary:**

Current masked diffusion models acceleration techniques use confident thresholds within a block. In order to increase efficiency, this requires lowering the confidence threshold. However, lowering the confidence threshold results in worse performance. This work argues that this degradation is due to the masked diffusion model capturing “unnecessary dependencies” between tokens.

To fix this, they propose interpreting masked diffusion inference as iterative token grouping, with the goal of grouping the largest possible set of tokens that can be sampled independently without losing coherence. They propose teaching a student model to reproduce the distributions obtained from two steps of a teacher model. They also propose using JSD (Jensen Shannon divergence) at inference to identify which positions are most dependent on revealed positions (and thereby less dependent on unmasked positions).

**Strengths:**

The conceptual motivation does make sense — I think this paper does articulate a fundamental problem with masked diffusion models that limit their efficiency.

**Weaknesses:**

**Unfair Baseline**:
The distillation requires training a LoRA adaptor on the GSM8K training set. However, fast-dLLM is a training free method. Comparing the two against each other feels like an apples-to-orange comparison — how can we compare an inference-time algorithm to a distillation algorithm? Both seem like orthogonal directions — can’t fast-dLLM also be applied to the distilled model? A more fair comparison would be against other distillation algorithms, like SDTT [1].


**Missing Related Work**: I did not see any citation or reference to Self-Distillation Through Time (SDTT). Given that the self-distillation proposed in the submission and in [1] are similar, I think this is a major omission. In fact, both algorithms seem extremely similar and I am unsure as to what the differences are.

**Algorithmic Limitation**:
I am not sure I quite understand how the distillation algorithm relates to disentangling token dependencies. It seems that the “confident” positions are identified using a confidence threshold, and then the target for the less confident positions are obtained by using a second forward pass, but with the confident positions sampled.

Why is model confidence related to how different tokens relate to each other? I don’t see how the relation between different tokens can be assessed or measured just by looking at confidence values. For example, take the following sentence:

“Manhattan is in [MASK] [MASK] …”
Here, both “New” and “York” could be very high confidence, but on their own they wouldn’t make much sense.

**Empirical Performance**:
The empirical performance does not seem to demonstrate a very clear advantage over fast-dLLM. The only task where the proposed method (distillation, no JSD) wins is the Math set — but this can be attributed to the model being fine-tuned via the distillation strategy on a math reasoning dataset. On all other datasets, the performance gain does not seem that significant — no increase in accuracy or speedup.

Also, it seems that JSD only provides benefits over distillation alone on GSM8K, and even then the benefits are minor. On the other datasets, distillation w/o JSD and with JSD seem to have similar performances.

[1] Beyond Autoregression: Fast LLMs via Self-Distillation Through Time. Deschenaux, Gulcehre. ICLR 2025.

**Questions:**

- How does confidence relate to token codependence? Is there any empirical justification or theoretical justification that shows that confident positions are more connected to less confident positions than other confident positions?
- How does the proposed self distillation compare to SDTT from [1]?
- How does the inference-only algorithm perform on the base model without self distillation?

---

### Official Review · Reviewer_WdYa · 2025-11-04

**Soundness:** 2
**Presentation:** 1
**Contribution:** 2
**Rating:** 2
**Confidence:** 4

**Summary:**

This paper works on diffusion language models and proposes two new methods for more efficient generation, specifically better parallel decoding. Their overall framework is called "Disentangled Decoding". First, the authors introduce a distillation strategy, "Dependency-Aware Self-Distillation", in which a teacher model generates first confident, and then the less confident tokens in two steps, and the student in trained to directly predict this two-step generation in a single step. Moreover, the authors argue that tokens that are determined primarily via already given context, and do not depend on other still masked token, can be unmasked and generated first. Based on this observation the paper proposes to measure during inference the Jenson Shannon Divergence (JSD) between the denoising distributions with and without prior context being masked -- tokens with high JSD then depend primarily on context, and therefore they can be generated more quickly, with different confidence thresholds than the other tokens. They call this method "Slow-Fast Decoding". Overall, the paper suggests to view the slow denoising of diffusion language models through the lens of token correlations: Accurate parallel and fast decoding is only possible when there are no dependencies among the tokens that are being unmasked. This principle is leveraged in the JSD-based decoding approach, for instance. The paper runs experimental validation on language generation tasks, including maths and coding. The methods perform on par with Fast-dLLM and outperforms naive greedy decoding where only one token per step is unmasked.

**Strengths:**

**Better than greedy baseline:** The numerical results demonstrate that compared to naive greedy decoding where only a single token is unmasked in every generation step, the proposed approach is superior. This is, it is much faster while preserving similar generation accuracy.

**Originality:** The idea to use the Jenson Shannon Divergence to estimate token dependency on given context, and decide the decoding strategy based on that, is novel and represents a creative idea.

**Weaknesses:**

There are various concerns:
- **Relation to Self-Distillation Through Time:** The idea to distill a multi-step teacher into a single-step student has been explored before in diffusion language models, in Self-Distillation Through Time (SDTT) [1]. The proposed Dependency-Aware Self-Distillation is related to SDTT. Hence, SDTT should be cited and discussed, and a comparison would be appropriate. However, SDTT is not even cited.
- **Dependency-Aware Self-Distillation motivation:** When explaining and motivating the Dependency-Aware Self-Distillation in the first paragraph in 4.1, the authors give the example with "financial" and "global" and say *"However, a standard MDM might learn an overly sensitive conditional model where predicting "global" is difficult until "financial" is revealed."*. The authors argue that an MDM may learn spurious, undesired token dependencies. Why would it learn any "incorrect" dependencies? This is not well motivated. I would like the authors to explain this better and I would also suggest to find actual examples in a trained MDM where this happens, i.e. where the model learnt undesired dependencies (instead of coming up with this artificial example in the first paragraph of 4.1). Currently, the argument made by the authors is weak.
- **Student behavior explanation in Dependency-Aware Self-Distillation:** Related, the explanation below equation (8) is also unclear (line 243-245): Yes, the student learns to predict the tokens in K in parallel, but these tokens were the ones that were initially decoded with high confidence by the teacher, thereby implying that these were actually tokens with little dependency. This contradicts the statement that the student prunes unnecessary dependencies -- for the tokens in K, there were already few dependencies, which is what allowed the teacher to unmask them first in the first place. This needs clarification.
- **Contradictory JSD explanations:** In line 182 and following, the authors write that tokens with high JSD are context-stable and can be decoded in parallel, and fast, and that low JSD tokens are ambiguous and should be decoded slowly and sequentially. Later, in line 317 and following, the authors instead write that tokens with low JSD go into the fast set and are decoded quickly, while high JSD tokens go into the slow set. These statements are contradictory. I would also suggest the authors to include an algorithm box, explaining the detailed JSD-based decoding recipe.
- **Presentation of numerical results:** The paper does never explicitly write what exactly the numbers in Table 1 and table 2 are, i.e. that these are reconstruction accuracies, and probably tokens/second as well as the relative speed-up compared to greedy decoding.
- **Weak numerical results:** The numerical evaluations show that the proposed methods outperform the greedy baseline, but they are not significantly better than the Fast-dLLM baseline in all experiments, and other comparisons are missing.
- **No MATH in Table 2:** Why are the authors not showing the MATH (4-shot) results in Table 2 with JSD, but in Table1? This is suspicious.
- **Evaluation metrics:** Generally, it would be good to also measure perplexity and diversity (e.g. via entropy), including in unconditional generation settings, and compare to baselines. These are common and relevant metrics.
- **Figure 2:** It seems Figure 2 is not explicitly referenced or discussed in the main text.
- **Missing Appendix Results:** The authors write *To mitigate this limitation, we extend our distillation training to other domains, and those results are reported in the Appendix* in line 382. But there are no such additional results in the appendix.

**Conclusions:** Due to the many concerns above, the paper does not meet the bar for acceptance.


[1] Deschenaux and Gulcehre, "Beyond Autoregression: Fast LLMs via Self-Distillation Through Time", ICLR 2025.

**Questions:**

Some questions:
- Why did the authors choose the $1/t$ weighting in the distillation loss in equation (8)? This was not well motivated.
- In Figure 1, why does the Spearman correlation coefficient $\rho$ decrease for later blocks (higher Block index)?

I have also various suggestions for improvements of the presentation and some typos:
- The citation style in this paper does not follow best practice. When a citation is not part of the sentence itself, it should be done in parentheses, as it is done in all papers in the field. The authors do not follow this style.
- Equation (2) is confusing: The left hand side seems to describe the distribution only over the fully denoised tokens $x_{B_i}$ for block $B_i$, whereas the right hand side seems to describe the product over all noise levels.
- In line 307, are the authors referring to Figure 1, instead of Figure 2?
- Line 333: Typo: o -> To.

---

### Note · Authors · 2025-11-29

I have read and agree with the venue's withdrawal policy on behalf of myself and my co-authors.